



# Throughfall isotopic composition in relation to drop size at the intra-event scale in a Mediterranean Scots pine stand

Juan Pinos[1], Jérôme Latron[1], Kazuki Nanko[2], Delphis F. Levia[3] and Pilar Llorens[1]

[1]Surface Hydrology and Erosion group, Department of Geosciences, IDAEA-CSIC, Barcelona, Spain
[2]Department of Disaster Prevention, Meteorology and Hydrology, Forestry and Forest Products Research Institute, Tsukuba, Japan
[3]Departments of Geography & Spatial Sciences and Plant & Soil Sciences, University of Delaware, Newark, DE, USA

*Correspondence to*: Pilar Llorens (pilar.llorens@idaea.csic.es)

**Abstract.** The major fraction of water reaching the forest floor is throughfall, which consists of free throughfall, splash throughfall and canopy drip. Research has shown that forest canopies modify the isotopic composition of throughfall by means of evaporation, isotopic exchange, canopy selection and mixing of rainfall waters. However, the effects of these factors in relation to throughfall isotopic composition and the throughfall drop size reaching the soil surface are unclear. Based on research in a mountainous Scots pine stand in northeastern Spain, this study sought to fill this knowledge gap by

examining the isotopic composition of throughfall in relation to throughfall drop size. In the experimental stand, throughfall consisted on average of 65% canopy drip, 19% free throughfall and 16% splash throughfall. The dynamics of the isotopic composition of throughfall and rainfall showed complex behavior throughout events. The isotopic shift showed no direct relationship with meteorological variables, number of drops, drop velocities, throughfall and rainfall amount, or raindrop kinetic energy. However, the experiment did reveal that the isotopic shift was higher at the beginning of an event, decreasing

as cumulative rainfall increased, and that it also increased when the median volume drop size of throughfall ($D_{50\_TF}$) approached or was lower than the median volume drop size of rainfall ($D_{50\_RF}$). This finding indicates that the major contribution of splash throughfall at the initial phase of rain events matched the highest vapor pressure deficit (VPD), and at the same time corresponded with higher isotopic enrichment, which implies that splash droplet evaporation occurred. Future applications of our approach will improve understanding of how throughfall isotopic composition may vary with drop type

and size during rainfall events across a range of forest types.

## 1 Introduction

Forests play an important role in the water balance of catchments by redistributing rainfall in throughfall, stemflow and
interception loss. To study the rainfall partitioning process, the classical hydrometric approach of measuring rainfall partitioning has been recently complemented and expanded by natural tracing with water stable isotopes ($\delta^{18}O$ and $\delta^2H$). It has been shown that the forest canopy modifies the isotopic composition of throughfall and stemflow in relation to open





rainfall (Allen et al., 2017; Cayuela et al., 2018a). Isotopic fractionation can occur in both directions (enrichment and depletion), with enrichment being more frequent (Saxena, 1986). Isotopic shifts are caused by four factors: evaporation,
isotopic exchange, canopy selection and mixing of waters (Allen et al., 2017), but the effect of each factor and the magnitude of the isotopic shift remain unclear. Isotopic fractionation by evaporation occurs when rain water molecules achieve enough energy to change from liquid to the gas phase, resulting in an enrichment of heavy isotopes. Isotopic exchange is the exchange between liquid and environmental vapor when these pools are not at an isotopic steady state. Canopy selection is the result of selective water retention in the canopy of different lapses within rainfall events that temporally vary its isotopic
composition. Mixing of water relates to the storage of the residual water of previous rainfall in the canopy that is eventually mixed with new rain water. Exchange, canopy selection and mixing of water can cause either isotopic enrichment or depletion.

Because throughfall represents the main water input to the soil (Levia and Frost, 2006), understanding the spatiotemporal
variability of throughfall isotopic composition is of paramount importance to use it as an input value in isotope-based hydrological studies. Spatial variability of the throughfall isotopic composition between collectors seems to be related to canopy cover (Cayuela et al., 2018a) but not to throughfall amount (Allen et al., 2015). However, all isotopic fractionation factors could very well occur during the same rainfall event, which complicates the understanding of the mechanisms that influence the intra-event isotopic differences between rainfall and throughfall. Although a small number of studies have
focused on understanding the spatiotemporal variability of throughfall isotopic composition at the intra-event scale (e.g., Kubota and Tsuboyama, 2003; Ikawa et al., 2011; Cayuela et al., 2018a), the factors controlling this variability remain largely unclear.

Laboratory experiments demonstrated that falling water droplets experience isotopic fractionation due to evaporation and
isotopic exchange with the environment; and that the degree of evaporation is related to drop characteristics (size, velocity, number, temperature), air conditions and exposure time or falling distance (Friedman et al., 1962; Stewart, 1975). More recently, Murakami (2006) and Dunkerley (2009) analyzed the concept of splash droplet evaporation, showing that numerous small droplets are produced when a raindrop hits the canopy, enhancing the evaporation of the droplets. However, the influence that splash generation and subsequent evaporation or ionic exchange exerts on the isotopic composition of
throughfall remains unexplored (Allen et al., 2017).

An increasing number of studies of throughfall drop characteristics, such as drop size, velocity and kinetic energy, have shed light on the partitioning of throughfall by trees. Some recent studies have shown that the way in which water reaches the forest floor is affected by throughfall drop characteristics and, therefore, affects soil erosion and probably soil moisture
(Levia et al., 2017; Nanko et al., 2020). Moreover, several studies showed that biotic and abiotic factors affect throughfall drop characteristics. These diverse factors have been related to whether trees are coniferous or broadleaved deciduous (Levia





et al., 2019), the presence or absence of foliage (Nanko et al., 2016), canopy species and meteorological factors (wind and rainfall intensity) (Nanko et al., 2006; Lüpke et al., 2019), physical leaf characteristics (Nanko et al., 2013), the thickness and saturation of the canopy (Nanko et al., 2008a) and the spatial variation between crown positions under a single tree (Nanko et al., 2011) or within tree stands (Nanko et al., 2020).


For a given rainfall event with simultaneous measurements of drop size distributions (DSDs) measured both inside and outside the forest, throughfall can be divided into three types: (1) free throughfall (FR), which is the proportion of throughfall that does not contact the canopy surface, thus maintaining the same DSD as open rainfall; (2) splash throughfall (SP), corresponding to the drops that hit the canopy and split into smaller drops; and (3) canopy drip (DP), which is the proportion of throughfall that is initially retained and routed by vegetative surfaces but eventually detaches from the vegetation (Levia et al., 2017; 2019). Canopy drip has the largest drop diameter and splash has the smallest (Levia et al., 2017).


Despite the important progress made in investigating throughfall dynamics using drop size data from disdrometers, the inter-relationships between throughfall isotopic composition and throughfall drop size need to be investigated at the intra-event scale to yield insights on evaporative demand (Allen et al., 2017; Cayuela et al., 2018a; Levia et al., 2011, 2017). To our knowledge, there are no studies analyzing the details of the interplay between fine-scale rainfall and throughfall drop characteristics in terms of isotopic composition. Accordingly, the specific objectives of this study were: (i) the quantification and analysis of isotopic composition and drop sizes of both rainfall and throughfall at the intra-event scale; (ii) the calculation of the proportion of each throughfall type (free throughfall, splash and drip); and (iii) the analysis of the inter-relationships of observed isotopic shift and drop size between open rainfall and throughfall.



## 2. Material and methods

### 2.1 Site description


The study was conducted in the Can Vila catchment (Fig. 1), one of the Vallcebre research catchments (NE Spain, 42° 12′N, 1° 49′E) in the eastern Pyrenees. These catchments have been monitored for 30 years for hydrological and ecohydrological purposes (see Latron et al., 2009; Llorens et al., 2018). Nowadays, most of the catchment is covered by Scot pine forests (*Pinus sylvestris L.*), which arose through afforestation of old agricultural terraces and small original fragmented oak forests (*Quercus pubescens Willd.*) (Poyatos et al., 2003). The climate is sub-Mediterranean with mean annual precipitation, reference evapotranspiration and air temperature of 867 ± 223 mm, 856 ± 69 mm and 9.2ºC, respectively (mean for the period 1999 to 2018). Precipitation is seasonal throughout the year, with spring and autumn the wettest seasons and summer and winter the driest ones. Evapotranspiration shows a seasonal pattern with maximum values in summer of up to 6.9






mm·day$^{-1}$. Our study is based on data obtained within the Scots pine stand of Cal Rotes, located in the central part of the Can
Vila catchment. The stand has an area of 900 m$^2$ and is located at 1,200 m elevation with a northeast aspect. The stand has a
mean diameter at breast height (DBH, 1.3 m) of 19.9 ± 9.2 cm, a stand density of 1,189 trees·ha$^{-1}$, a stand basal area of 45.1
m$^2$·ha$^{-1}$, a mean tree height of 17 ± 4.4 m and a mean canopy cover of 69.3 ± 17.7% (Molina et al., 2019).

## 2.2. Monitoring design and data collection

The experimental work involved the continuous measurement, characterization and sampling of open rainfall and
throughfall. The rainfall monitoring site was located in an open area approximately 100 m from the Scots pine stand where
throughfall was monitored (Fig. 1) at two different distances (0.82 and 1.15 m) from the bole of the same tree (Table 1).
Other nearby trees that might affect the throughfall monitoring location were located at an average distance of 4.4 ± 1.1 m.

The monitoring design used ground-based laser disdrometers developed by Nanko et al. (2006; 2008b) (see *Laser
disdrometer characteristics* section), one for open rainfall and two for throughfall, each placed just above a tipping-bucket
rain gauge (model AW-P, Institut Analític) with 0.2 mm resolution. The rainfall and throughfall passing through the laser
disdrometers and the tipping-buckets were sequentially collected at 5 mm rainfall intervals (i.e. both samplers switched to
the next bottle simultaneously) by means of automatic samplers (ISCO 3700) buried in the ground to prevent evaporation
from the water samples (Fig. 2). According to Iida et al. (2020), dynamic calibration of the tipping-buckets was performed to
ensure the quality of data. The tipping-buckets and automatic samplers were connected to a datalogger (DT85, Datataker)
that recorded cumulative rainfall/throughfall amounts every 5 minutes. Unfortunately, the disdrometer located furthest from
the tree lost a substantial amount of throughfall data due to technical problems and was therefore discarded from the
analysis.


### 2.2.1. Laser disdrometer characteristics

The laser disdrometers continuously measured the number of drops, as well as individual drop size and velocity. The
instruments were built using a laser transmitter and a receiver (IB-30, KEYENCE Corporation) with an amplifier (IB-1000,
KEYENCE Corporation) covered by two protection screens, following the design by Nanko et al. (2006). The sensors were
attached to an iron frame. The light source of the laser sensor was a visible semiconductor laser of 660 nm. Drops were
measured within a 4,500 mm$^2$ sampling area (30 mm wide and 150 mm long) of 1 mm thickness. When a drop passed
through the laser beam, the receiving laser beam decreased and the output voltage from the amplifier fell in proportion to the
intercepted area of the laser beam. The output voltage was collected by an Arduino UNO every 50 micro seconds (= 20kHz).
The output voltage data was converted into drop diameter and velocity data. The detailed calculation protocol is shown in
Nanko et al. (2020). The shape of raindrops was assumed to be an oblate spheroid, whose axis ratio was determined by





Andsager et al. (1999). The recorded drop data were collected weekly (emptying the Arduino SD memory card) and later post-processed at 5-min intervals, arranged in 0.1 mm drop size classes, with their respective numbers of drops and drop velocities computed. The Arduino datalogging system used in this study had some limitations, as it could not record all the drops passing simultaneously through the laser beam (see Appendix A for further details).


### 2.2.2 Meteorological data

Meteorological data were obtained from an automatic weather station located 2 m above the canopy of the forest stand. The station was equipped with the following sensors: an air temperature and relative humidity probe (HMP45C, Vaisala), an anemometer and wind vane (A100R, Vector Instruments) and a net radiometer (NR Lite, Kipp & Zonen). Data were
measured every 30 s and averaged at 5-min intervals with a datalogger (DT85 Datataker).

### 2.2.3 Event classification

This study was carried out on an event basis. A rain-free period of 6 h (day) and 12 h (night), allowing for the drying of the canopy, was considered necessary to define separated events (Llorens et al., 2014). All event data were evaluated (i.e.
quality-controlled) for potential errors, and events with missing or erratic data were discarded. The definition of rainfall event classes was based on the duration and intensity of the event according to the following criteria: (a) rainfall duration of 7 h was used to distinguish between short and long rainfall events; and (b) a maximum 30-min rainfall intensity threshold of 10 mm h$^{-1}$ was used to separate low- and high-intensity events. By using both thresholds, rainfall was classified as: (1) short duration-low intensity (S-L) ($\leq$ 7 h and $\leq$ 10 mm h$^{-1}$); (2) short duration-high intensity (S-H) ($\leq$ 7 h and > 10 mm h$^{-1}$); (3)
long duration-low intensity (L-L) (> 7 h and $\leq$ 10 mm h$^{-1}$); and (4) long duration-high intensity (L-H) (> 7 h and > 10 mm h$^{-1}$).

### 2.3 Estimation of throughfall types

The simultaneous measured open rainfall and throughfall DSD data were used for the separation of throughfall types by
applying the protocol described by Levia et al. (2019). The separation, based on the DSD of throughfall and rainfall, consists of the calculation of the accumulated volume for each 0.1 mm drop diameter class. For each class $i$ the volume of throughfall ($TF_i$) is partitioned in the corresponding class $i$ of free throughfall ($FR_i$), splash throughfall ($SP_i$) and canopy drip ($DP_i$).

$$\Sigma TF_i = \Sigma(FR_i + SP_i + DP_i) \tag{1}$$

$FR_i$ is calculated as:



$$FR_i = p\, OP_i \qquad (2)$$

where $p$ is the maximum value under the condition ($FR_i$ - $p\, OP_i$) > 0.

Drops, with a diameter ($d_i$) < 1 mm, were considered as splash throughfall; and the maximum splash throughfall diameter ($D_{MAX\_SP}$) was set at 2 mm. A Weibull cumulative distribution function (Eq. 3) was used to determine the distribution of $SP_i$ between 1 mm and $D_{MAX\_SP}$. In this study, the minimum splash drop diameter was set at 0.5 mm [(rather than the 0.4 mm by Levia et al. (2019)], since the datalogging systems were different: Arduino in this study and laptop in Levia et al. (2019). Therefore, in the Weibull function this value was set at 0.5 instead of 0.4.

$$F(d_i) = 1 - exp\left\{-\left(\frac{d_i - 0.5}{b}\right)^c\right\} \qquad (3)$$

Equation 4 was used for the calculation of the estimated splash throughfall distribution ($SP^*_i$).

$$SP^*_i = \{\Sigma(TF_i - FR_i)\}\{F(d_i) - F(d_{i-1})\} \qquad (4)$$

$SP_i$ is determined by the minimum value between $SP^*_i$ and ($TF_i$ - $FR_i$). Finally, $DR_i$ was calculated using Eq. 5 when $d_i$ > $D_{MAX\_SP}$ or by Eq. 6 when splash was present.

$$DR_i = TF_i - FR_i \qquad (5)$$
$$DR_i = TF_i - FR_i - SP_i \qquad (6)$$

For a detailed explanation of the formulas, calculations and assumptions employed, the reader is referred to Levia et al. (2017, 2019).

**2.4 Isotopic analysis**

Rainfall and throughfall samples collected by the automatic samplers were analyzed for water stable isotopes ($\delta^{18}O$ and $\delta^2H$) by the Scientific-Technical Services of the University of Lleida using the Cavity Ring-Down Spectroscopy technique with a Picarro L2120-i analyzer (Picarro Inc.). The equipment had an accuracy of < 0.1‰ for $\delta^{18}O$ and < 0.4‰ for $\delta^2H$, based on the repetition of four reference samples provided by the International Atomic Energy Agency (IAEA).

All isotopic data were expressed in terms of $\delta$ values and calculated as:





$$\delta = \left(\frac{R_{sample}}{R_{VSMOW}} - 1\right) \cdot 1000\text{\textperthousand} \tag{7}$$

where *VSMOW* is the Vienna Standard Mean Ocean Water and *R* is the isotope ratio ($^{18}O/^{16}O$ or $^{2}H/^{1}H$). The isotopic shift between throughfall and open rainfall ($\Delta\delta^{18}O_{TF-RF}$) corresponds to the direct difference between the values of $\delta^{18}O$ throughfall and $\delta^{18}O$ open rainfall:

$$\Delta\delta^{18}O_{TF-RF} = \delta^{18}O_{TF} - \delta^{18}O_{RF} \tag{8}$$

Deuterium excess (*d*-excess) was later determined to describe the deviation from the meteoric water line (MWL) and to indicate kinetic fractionation effects caused by evaporation, as in Gat (1996):

$$d\ excess = \delta^{2}H - 8 \cdot \delta^{18}O \tag{9}$$

**2.5 Statistical analysis**

IBM SPSS Statistics 25 software (IBM Corporation) was employed for the statistical analyses. As the correlation between variables of our dataset was not necessarily linear, the Spearman's rank correlation coefficient ($R_s$) was computed. Data not normally distributed were analyzed by the non-parametric rank-based Kruskall-Wallis *H* test, which examines the significance of the differences among throughfall type percentages or drop diameters with respect to the grouping of the four rainfall classes (based on duration and intensity). Statistical significance was set at $p < 0.05$. If the *H*-value from the Kruskal-

Wallis test was significant, the Mann-Whitney-Wilcoxon was applied as a *post hoc* test for the pairwise comparisons to determine which groups were significantly different.

**3 Results and Discussion**

**3.1 Open rainfall, throughfall and drop characteristics**

Twenty-one rainfall events were selected for analysis during the observation period (May 2018 to July 2019) (Table S1), amounting to a total rainfall of 482 mm. The rainfall depth per event ranged from 6.0 to 52.5 mm and the maximum 30-min intensities varied between 2.7 and 38.2 mm h$^{-1}$. The total amount of throughfall for the selected events was 428 mm, equivalent to 89% of total incident rainfall, and the maximum throughfall was 48.3 mm. The total amount of

rainfall/throughfall collected in the 21 events was distributed in 98 pairs of samples collected at 5 mm rainfall intervals. For 33 of the 98 pairs of samples, throughfall was higher than rainfall; one third of these samples corresponded to the end of the rainfall event, after rainfall stopped, while the remaining two-thirds were distributed without any specific pattern at different time intervals during the rainfall events.





The total number of drops in the dataset (i.e., the 98 samples) was 529,750 for open rainfall and 271,963 for throughfall,
which means that the number of throughfall drops was 48% lower than of rainfall ones. Altogether, 88% of the samples had
fewer throughfall drops than rainfall (Fig. S1a). The median volume drop diameter ($D_{50}$), calculated for the 98 pairs of
samples, ranged between 1.20 and 4.44 mm for open rainfall, and between 1.47 and 4.17 mm for throughfall. The maximum
diameter ($D_{MAX}$) ranged between 2.51 and 7.87 mm for open rainfall and between 3.25 and 7.92 mm for throughfall. At the

event scale, the median volume drop diameter ($D_{50}$) for open rainfall ranged between 1.36 and 3.24 mm, and for throughfall
between 2.83 and 3.90 mm. Overall, the mean throughfall $D_{50}$ found in this study (3.36 mm) was larger than those reported
in other DSD studies. For example, Nanko et al. (2006) found that the throughfall $D_{50}$ ranged from 1.77 to 2.93 mm for two
coniferous species (Japanese cypress and Japanese cedar) in different meteorological conditions; and Lüpke et al. (2019)
reported throughfall $D_{50}$ values of 2.7 and 0.80 for a European beech and Norway spruce tree. The throughfall $D_{50}$ was on

average 1.3 mm larger than the rainfall $D_{50}$. However, in moments with very large rainfall drops, generally during the first
two rainfall intervals (i.e. $\leq$ 10 mm), rainfall (8% of the total samples) had on average a diameter 0.37 mm larger than
throughfall ones (Fig. S1b). Mean drop velocity was $4.34 \pm 0.62$ m·s$^{-1}$ for rainfall and $3.97 \pm 0.29$ m·s$^{-1}$ for throughfall. As
expected, the mean velocity of throughfall drops was on average slower (0.4 m·s$^{-1}$) than rainfall (Fig. S1c), due to the
differences in drop falling distance caused by the canopy.


**3.2 Partitioning throughfall types**

Canopy drip, free throughfall and splash throughfall represented respectively 65%, 19% and 16% of the total throughfall
volume collected (Fig. 3). In comparison with our results, Levia et al. (2019) found less canopy drip (51%), but higher free
throughfall (31%), and a similar splash percentage (18%) for other types of coniferous species. Tree height and canopy

architecture differences between the coniferous species investigated by Levia et al. (2019) and the trees in our study may
explain the differences in throughfall type percentages. In our study plot, higher tree canopy density, together with more
woody surfaces (branches that may be dying or shed) from the lower part of the crown towards the stem base, probably
reduced the contribution of free throughfall but raised canopy drip, in comparison with the shorter coniferous trees
considered by Levia et al. (2019).


When separating events by rainfall classes (depending on rainfall duration and intensity), the Kruskal-Wallis test indicated
that the percentages of splash throughfall were not significantly different between classes ($H = 3.34, p = 0.342$). In contrast,
the percentages between classes for free throughfall ($H = 12.22, p = 0.007$) and canopy drip ($H = 15.16, p = 0.002$) were
significantly different. The Mann-Whitney-Wilcoxon *post hoc* pairwise comparisons indicated that long duration-low

intensity rainfall events had a significantly lower percentage of free throughfall and higher percentage of canopy drip than
long duration-high intensity events ($p = 0.009$ for both) and short duration-high intensity ones ($p = 0.004$ and 0.002,





respectively). Further, short duration-low intensity events had a significantly higher percentage of canopy drip than short duration-high intensity events ($p = 0.004$).

The median volume drop diameters of the canopy drip, free throughfall and splash throughfall averaged for the 21 studied events were 4.28, 2.12 and 1.36 mm, respectively (Fig. 3). When analyzing drop diameters in cumulative drop volume percentiles, the Kruskal-Wallis test showed that drop diameters in the four rainfall classes were not significantly different for splash throughfall ($H$ ranging from 1.12 to 2.44 and $p$ ranging from 0.478 to 0.772) and canopy drip ($H$ ranging from 1.12 to 7.46 and $p$ ranging from 0.059 to 0.773), with the exception of the 75th percentile which was significantly different for 255 canopy drip ($H = 9.36$, $p = 0.025$). The Mann-Whitney-Wilcoxon *post hoc* pairwise comparisons indicated that short duration-low intensity rainfall events had significantly smaller canopy drip than short duration-high intensity ones ($p = 0.017$). As expected, the Kruskal-Wallis test revealed that the free throughfall diameter was significantly different in the four rainfall classes ($H$ ranging from 13.22 to 14.52 and $p$ ranging from 0.002 to 0.004).

In summary, throughfall during low intensity events gave higher canopy drip percentages (69% and 72%, for S-L and L-L events, respectively) and lower free throughfall (16% and 13%, respectively) than events with high intensities (56% and 62% of canopy drip and 25% and 21% of free throughfall for S-H and L-H, respectively). Short duration-low intensity events generated smaller canopy drip diameters ($D_{50\_DR} = 4.03$ mm); and long duration-low intensity events, smaller free throughfall drop diameters ($D_{50\_FR} = 1.52$ mm). Based on volume, our results show that low rainfall intensities increased 265 canopy drip in both short and long events. On the other hand, rainfall duration increased canopy drip for both low- and high-intensity events. Therefore, long rainfall events with low rainfall intensity yielded the highest percentage of drip, whereas short rainfall events with high rainfall intensities yielded the lowest (difference of 16%).

**3.3 Isotopic composition of open rainfall and throughfall**

The $\delta^{18}$O isotopic composition of the 98 pair samples (21 studied events) ranged from -13.72 to -2.18‰ for open rainfall and from -13.65 to -2.20‰ for throughfall. For $\delta^2$H, values ranged from -101.25 to -4.84‰ for open rainfall and from -98.54 to -3.61‰ for throughfall. As shown in Fig. 4a, open rainfall and throughfall samples fell on the local meteoric water line (LMWL) defined for the Vallcebre catchments ($\delta^2$H = 7.9 $\delta^{18}$O + 12.9) (Casellas et al., 2019). Most of the throughfall samples (83%) were more enriched ($\delta^{18}$O and $\delta^2$H) than the open rainfall ones, showing predominant enrichment rather than 275 depletion, which corroborates the results of several previous studies (Saxena, 1986; Dewalle and Swistock, 1994; Kubota and Tsuboyama, 2003; Cayuela et al., 2018a). The isotopic shift between throughfall and open rainfall ($\Delta\delta^{18}$O$_{TF-RF}$) ranged from -1.48‰ to 2.17‰. These differences are slightly higher than those reported by Cayuela et al. (2018a) for the same stand, probably due to the difference in the number of throughfall collectors in the two studies (10 *vs* 1 in our study). Our results indicated preferential throughfall enrichment at the event scale, based on the volume-weighted mean of $\delta^{18}$O (Table





S1), contrary to those from Xu et al. (2014) who reported preferential throughfall depletion for a *Pinus radiata* forest in a
Mediterranean climate. However, the values reported by these authors were bulked over multiple events. This highlights the
paramount importance of using finer-scale sampling resolutions. Figure 4b indicates the presence of non-equilibrium
fractionation processes, since not all the enriched samples corresponded with a decrease in deuterium excess; and not all the
depleted samples, with an increase. In fact, 50% of the $\delta^{18}O$ enriched samples of throughfall had negative deuterium excess
difference. Similarly, Herbstritt et al. (2019) observed that enrichment does not always lead to negative deuterium excess
values and argued that such a phenomenon is usually attributed to mixing processes. For the case of pre-event water mixing
(Allen et al., 2014), we ensured that all measured events started with an initially dry canopy, preventing the mixing of event
water with water stored previously in the canopy. Consequently, it remains unclear to what extent the differences in isotopic
composition between rainfall and throughfall can be attributed to mixing processes.


### 3.4 Drop sizes, throughfall types and isotopic composition in rainfall events of different durations and intensities

To improve understanding of the temporal dynamics of throughfall types, the drop diameter of canopy drip and the isotopic
composition of rainfall and throughfall, four events representative of each rainfall class were investigated in detail (i.e. at 5-
min intervals, Fig. 5). The main characteristics of these events that occurred in spring 2018 and 2019 are shown in Table 2.
Rainfall classes grouped by intensities showed similar intensity values (i.e. S-L with L-L, and S-H with L-H), whereas when
grouped by duration the time values were almost double (i.e. S-L with S-H, and L-L with L-H). Throughfall amount was
lower than incident rainfall except for the event on 11 June 2019 (event c in Fig. 5), in which it was slightly higher.

For the short duration-low intensity event (S-L, Fig. 5a), throughfall was mainly composed of free throughfall and splash
throughfall during the first 30 min (< 0.6 mm of rain) and the canopy drip diameter ($D_{50\_DR}$) was almost constant with a
mean of 1.54 mm. After 30 min, the percentage of canopy drip gradually increased for 2 hours, as well as the drop diameter
that reached an average $D_{50\_DR}$ of 3.65 mm, with a maximum canopy drip diameter ($D_{MAX\_DR}$) of 4.70 mm. Canopy drip is
clearly the main throughfall type during the last 30 min of the event, but the average $D_{50\_DR}$ decreased to 3.17 mm. Levia et
al. (2019) observed a similar trend for the proportionate contribution of throughfall types for coniferous trees in a simulated
steady event of short duration. As the water corresponding to the first ~1.9 mm of rainfall (1 h from the beginning of the
event) was mixed with pre-event water in the sampling bottle, it was discarded from the analysis. Throughfall isotopic
composition of the first sample was enriched (1.89‰) compared to open rainfall when splash throughfall reached 16.6%,
which may be related to increased evaporation. For the second sample, with a high contribution of canopy drip (75.8%), the
isotopic shift was almost zero, indicating a strong reduction of evaporation fractionation.


During the first 30 min (< 3.2 mm of rain) of the short duration-high intensity event (S-H, Fig. 5b) there was a gradual
decrease in splash throughfall, balanced by an increase in canopy drip, whereas free throughfall remained relatively stable.





During this time interval the $D_{50\_DR}$ increased from 1.60 to 4.33 mm. After 30 min, the contribution of various throughfall types was highly variable between successive time steps. Overall, canopy drip remained the main throughfall type during the event, but free throughfall tended to increase with rainfall intensity (from t = 1:00 to 1:30h), whereas splash throughfall increased from 13% (t = 0:30 to 2:40h) to 21% (t = 2:40 to the end of the event) as rainfall intensity decreased. During the central part of the event (t = 0:30 to 4:30h), the mean $D_{50\_DR}$ and $D_{MAX}$ were 4.04 and 5.48 mm, respectively; when rainfall almost stopped (t=4:30h), $D_{50\_DR}$ decreased to 2.33 mm. Similar to the previous event, isotopic composition of the first throughfall sample was more enriched (0.93‰) than open rainfall, with a splash contribution around 11.8%. Throughfall isotopic composition of the second sample was slightly enriched (0.35‰), probably as a consequence of the canopy drip increase (from 56 to 65%), even if splash throughfall type also increased to 14.5%. Enrichment of the third sample was similar to that of the first sample (0.92‰), with splash contribution of 12.6% and canopy drip contribution of 55%. The isotopic shift for the last two samples was almost zero. For these two samples, higher rainfall intensities may have reduced the lag time between throughfall and rainfall. Therefore, the evaporation impact may have led to periods of null or minimum isotopic shift, as suggested by Ikawa et al. (2011).

For the long duration-low intensity event (L-L, Fig. 5c), canopy drip was clearly the main throughfall type during the entire event with an average contribution of 80% (Table 2). For this very light rain during a long time period, the canopy probably intercepted almost all the raindrops, triggering canopy drip after the water storage capacity of the vegetative surfaces was exceeded. Data showed that the drop diameter broadly stabilized after 80 min with an average $D_{50\_DR}$ of 3.87 mm and $D_{MAX\_DR}$ of 5.10 mm until rainfall stopped for the first time (t = 11:30h). This intra-storm gap without rain was probably too short (< 90 min) to document any drying effect of the canopy on DSD, but a reduction in the canopy drop diameter at the beginning of the second burst of rainfall (t = 13:00h) was observed ($D_{50\_DR}$ of 2.58 mm). The throughfall isotopic composition was enriched for all the samples of the event. Since the splash throughfall type was small, splash droplet evaporation did not exert any significant influence. As mentioned, most of the rainwater was intercepted by the canopy and retained on the vegetative surfaces for large periods of time (between ~1 and ~6.5 h). Partial evaporation probably took place during these periods, which explains the isotopic enrichment of the throughfall samples (Xu et al., 2014).

Finally, intermittent rain showers in the long duration-high intensity event (L-H, Fig. 5d) produced a heterogeneous contribution of throughfall types during the entire rainfall event. Canopy drip was evidently the main type, free throughfall percentage increased with rainfall intensity and splash increased during low-intensity intervals. During a 20 min-long period (t = 8:10 to 8:35) without rainfall, throughfall was formed only by canopy drip and splash throughfall. This suggests that, in the absence of rainfall, dripping from the upper canopy may have impacted the lower canopy layers, which subsequently produced splash droplets, as also observed by Nanko et al. (2011). In this event, canopy drip increased after 10 min of rainfall, with mean $D_{50\_DR}$ rising from 2.16 mm to 4.28 mm and $D_{MAX\_DR}$ from 2.79 mm to 5.83 mm, until the end of the period with more intense rainfall (~ t = 5 h). Subsequently, the average $D_{50\_DR}$ decreased to 3.39 mm and $D_{MAX\_DR}$ to 4.38


mm. An unclear pattern in the isotopic shift between rainfall and throughfall was observed throughout the event, even if for most of the samples throughfall isotopic composition was only slightly enriched. This heterogeneous distribution of throughfall types and the absence of a clear pattern of isotopic shift suggest that, during this type of large event, a

combination of evaporation, isotopic exchange and canopy selection processes probably occurred, which was similar to the findings of Cayuela et al. (2018a).

In general terms, the dynamics of throughfall partitioning were different for each rainfall type, which caused changes in the proportions of throughfall type (data at 5-minute intervals) as well as in the isotopic composition (samples at 5 mm intervals)

at the finer scale within individual rain events. It seems that the increase or decrease in canopy drip diameter was related to rainfall event evolution, being lower at the beginning and the end of all events. Similar to Lüpke et al. (2019), splash throughfall for a coniferous species was important at the beginning of the analyzed events.

### 3.5 Relationship between isotopic shift and rainfall/throughfall characteristics

For the complete sample dataset (98 pairs of rainfall/throughfall samples), Spearman's rank-order correlation revealed no direct relationship between the $\Delta\delta^{18}O$ isotopic shift ($\Delta\delta^{18}O_{TF-RF}$) and meteorological variables such as VPD ($R_s = -0.075$, $p = 0.464$) or wind velocity ($R_s = -0.027$, $p = 0.795$) (Fig. S2a and b). This confirms the results of Herbstritt et al. (2019), who found that meteorological variables did not provide consistent evidence to explain the observed isotopic shift. The most likely scenario is that multiple factors/variables exerted influence on the isotopic fractionation observed in the canopy.

Moreover, Spearman's test showed no relationship between $\Delta\delta^{18}O_{TF-RF}$ and the difference in the number of drops between rainfall and throughfall ($R_s = -0.048$, $p = 0.642$), in drop velocities ($R_s = 0.114$, $p = 0.262$) or in amounts per sample ($R_s = -0.193$, $p = 0.057$) (Figs. S2c, d and e). However, some significant trends were observed. The isotopic shift ($\delta^{18}O$) between throughfall and rainfall increased when the $D_{50\_TF}$ got closer to or lower than the $D_{50\_RF}$ (i.e. with smaller throughfall drops) (Fig. 6a). The isotopic shift also decreased with increasing cumulative rainfall (average values shifted from 0.44 to 0.14‰

for rainfall between 5 and more than 40 mm) (Fig. 6b). This pattern is consistent with results found by Allen et al. (2017) in their meta-analysis showing larger isotopic shift differences for events with lower rainfall amounts. On the contrary, no relationship was found between the isotopic shift and the sampling time (the time each 5 mm sample took to be filled) (Fig. 6c), as shown by the relatively stable isotopic shift (on average 0.40‰) observed for sampling times ranging from fewer than 30 minutes to more than 8 hours. Finally, no clear relationship was found between the $\Delta\delta^{18}O$ isotopic shift and the kinetic

energy of the rainfall drops (Fig. 6d). Similar results were obtained for the $\delta^2H$ isotopic shift (data not shown). The variability observed in the isotopic shift was found to decrease with increasing cumulative rainfall and sampling times (Fig. S3a and b) and above a 300 J·m$^{-2}$ threshold for rainfall kinetic energy (Fig. 6d).





The intra-event dynamics of the isotopic shift between rainfall and throughfall were analyzed for events with a rainfall depth
larger than 10 mm (i.e. for events giving more than two water samples). A total of 88 water samples corresponding to 16
events, with a rainfall depth ranging from 12.5 mm to 52.5 mm, were selected. Following Cayuela et al. (2018a), the selected
events were split according to initial, middle and final stages. The initial stage corresponded to the first 5 mm of the event;
the middle stage consisted of all samples between the first and the last sample; and the final stage coincided with the sample
collected during the last 5 mm of the event.


The higher shift ($\Delta\delta^{18}O_{TF-RF}$) between throughfall and rainfall coincided with a higher VPD at the initial phase; and the lower
$\Delta\delta^{18}O_{TF-RF}$, with a lower VPD at the final phase of the events (Fig. 7). The first samples presented a median difference of
0.49‰ and, except for one outlier, all samples had positive $\Delta\delta^{18}O_{TF-RF}$, indicating throughfall enrichment. This isotopic
enrichment at the beginning of the events was congruent with the higher VPD observed, with a mean value of 0.12 kPa,
indicating higher atmospheric demand, which could increase evaporation in the canopy. Ikawa et al. (2011) and Cayuela et
al. (2018a) observed the same fractionation pattern and suggested a greater impact of evaporation at the beginning of the
event. Congruent with the isotopic shift and VPD dynamics during the events, the higher contribution of splash throughfall
(17%) also corresponded to the initial phase of events; and the lower splash contribution (14%), to the final phase of events
(Fig. 7). Although the difference between the initial and final phases seems small (3%), calculated percentages are based on
volume: to achieve this difference, a huge amount of splash droplets is required. Because splash droplets are prone to a high
degree of evaporation during their fall towards the ground (Dunin et al., 1988; Murakami, 2006; Xie et al., 2007), it is
inferred that the net contribution of splash throughfall based on volume is linked to the splash evaporation mechanism that
exerts influence at the initial stage of events, leading to greater isotopic enrichment of throughfall than of open rainfall.

On the other hand, the lower contribution of canopy drip (62%) corresponded to the initial phase of events; and higher drip
contribution (71%), to the final phase of events. Since larger drop sizes reduce droplet evaporation rates, as already
demonstrated several decades ago (e.g., Best, 1952; Brain and Butler, 1985), an increase in canopy drip contribution should
reduce the isotopic shift due to fractionation by evaporation. However, canopy drip may also be the result of water
accumulation originating from different flow paths (e.g., branchflow diverted from stemflow, drip recapture by lower canopy
layers), representing a mixing process over canopy surfaces (e.g., vertical redistribution of water can trigger mixing of water
from various small reservoirs formed by bark microrelief or cones in pine species), which may cause ambiguity in the
isotopic shift between throughfall and open rainfall. In addition, as mentioned by Herbstritt et al. (2019), the mechanistic
understanding of the variability of mixing between leaves (i.e. the water drip from leaf to leaf being able or not to cause
subsequent splashing) could also be a key element in water mixing and evaporation in the rainfall interception processes.

**4 Conclusions**





This study sought to measure the isotopic compositions and drop characteristics of both rainfall and throughfall at the intra-event scale and to examine if there is any correspondence between the rainfall-throughfall isotopic shift and their drop size distribution differences. Results showed that throughfall showed a lower number of drops, slower drop velocity and larger drop diameter than open rainfall did. Canopy drip accounts for most throughfall based on volume and corresponds to the largest drop diameter (average $D_{50\_DR}$ of 4.28 mm). Furthermore, our results showed that rainfall characteristics are an important abiotic factor that affects the throughfall DSD and consequently the proportion of throughfall types. Throughfall samples were almost always more enriched ($\delta^2H$ and $\delta^{18}O$) than rainfall. No correlation for the isotopic shift ($\delta^{18}O$ and $\delta^2H$) between throughfall and open rainfall was found in relation to meteorological variables, number of drops, drop velocities, throughfall and rainfall amount or raindrop kinetic energy. However, the experiment did reveal that the isotopic shift decreased during the progression of discrete rainfall events and increased with a larger proportion of splash droplets. Our key finding indicates that higher contribution of splash throughfall and higher VPD at the initial stage of the rainfall events correspond to a greater isotopic shift ($\Delta\delta^{18}O_{TF-RF}$). This provides evidence for the net contribution of splash droplets to isotopic enrichment by means of the greater evaporation of throughfall than of open rainfall.

Future research should aim to assess the intra-canopy mixing of waters during intra-event wetting/drying cycles of a rainfall event, to distinguish the isotopic fractionation factors. Additionally, using more throughfall tipping-buckets with disdrometers in different locations below the canopy would help to further evaluate the spatial variability of DSD and its relationship with isotopic composition. Future research should therefore focus on the use of fine spatiotemporal resolution of the isotopic composition of open rainfall and throughfall, together with meteorological variables and the various proportions of the different types of throughfall, to enable better understanding of the physical processes controlling differences in the isotopic shift in different tree species. Such an improvement in our understanding of the fine-scale mechanism of the isotopic composition of throughfall in relation to throughfall drop size would permit strengthen assumptions of forest-water interactions.

**Data availability.** Request for materials should be addressed to Pilar Llorens.

**Author contributions.** PL, JL, KN, and DL conceived the idea and designed the experiment. PL obtained funding. JP and JL collected the data. JP carried out the data analysis and wrote the initial draft of the paper. All authors discussed the results and edited the paper.

**Competing interests.** The authors declare that they have no conflict of interest.

**Acknowledgments.** We thank B. Singla for her relevant contribution to the preliminary phase of this study. We are also grateful to E. Sánchez, G. Bertran, C. Cayuela, A. Molina and F. Gallart for their support during the field work. We acknowledge support of the publication fee by the CSIC Open Access Publication Support Initiative through its Unit of Information Resources for Research (URICI).





**Financial support.** This research has been supported by the Spanish Ministry of Science and Innovation (project no. CGL2016-75957-R AEI/FEDER UE and grant no. BES-2017-082234) and the Japan Society for the Promotion of Science (JSPS KAKENHI grant no. JP15H05626).

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



**Tables**


**Table 1.** Biometric characteristics of the monitored tree (adapted from Cayuela et al., 2018b).

| | | | |
|---|---|---|---|
| Diameter at breast height (cm) | 35.2 | Crown volume ($m^3$) | 228 |
| Basal area ($cm^2$) | 973.1 | Mean branch angle (º) | 19.2 |
| Height (m) | 22.3 | Mean branch diameter (cm) | 4.4 |
| Canopy cover (%)[1] | 85.2 | Tree lean (º) | 7.9 |
| Crown area ($m^2$) | 17.3 | Distance to first live branch (m) | 12.4 |

[1] Canopy cover was measured over the throughfall tipping-bucket collection area.









**Table 2.** Measured variables for the four selected events. S-L: short duration-low intensity, S-H: short duration-high intensity, L-L: long duration-low intensity, L-H: long duration-high intensity. RF: rainfall, TF: throughfall, D: rainfall duration, $I_{Max}$: maximum 30-min rainfall intensity, N: number of drops, $D_{50}$: median volume drop diameter, SP: splash throughfall, FR: free throughfall, and DR: canopy drip.

| Date | RF class | RF (mm) | TF (mm) | D (h) | $I_{Max}$ (mm·h$^{-1}$) | $N_{RF}$ | $N_{TF}$ | $D_{50\_RF}$ | $D_{50\_TF}$ | SP (%) | FR (%) | DR (%) |
|---|---|---|---|---|---|---|---|---|---|---|---|---|
| 10 June 2018 | S-L | 7.8 | 5.4 | 3.0 | 5.9 | 9367 | 4230 | 1.9 | 3.4 | 17 | 16 | 67 |
| 06 June 2018 | S-H | 26.9 | 25.7 | 5.8 | 18.3 | 31472 | 19922 | 2.1 | 3.5 | 14 | 24 | 62 |
| 11 June 2019 | L-L | 32.6 | 33.0 | 16.2 | 5.9 | 58073 | 12753 | 1.4 | 3.9 | 9 | 11 | 80 |
| 12 May 2018 | L-H | 52.5 | 48.3 | 9.8 | 19.9 | 60559 | 32893 | 2.2 | 3.3 | 17 | 23 | 60 |



**Figures**

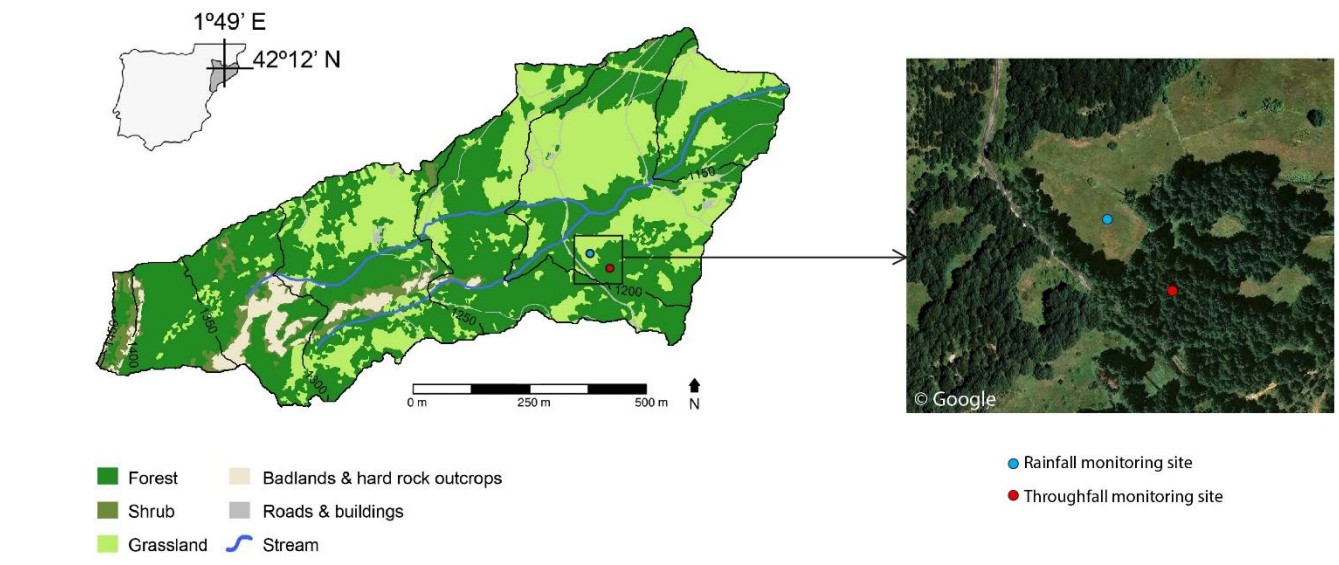


**Figure 1.** Location of the monitoring sites within the Can Vila catchment, Spain.









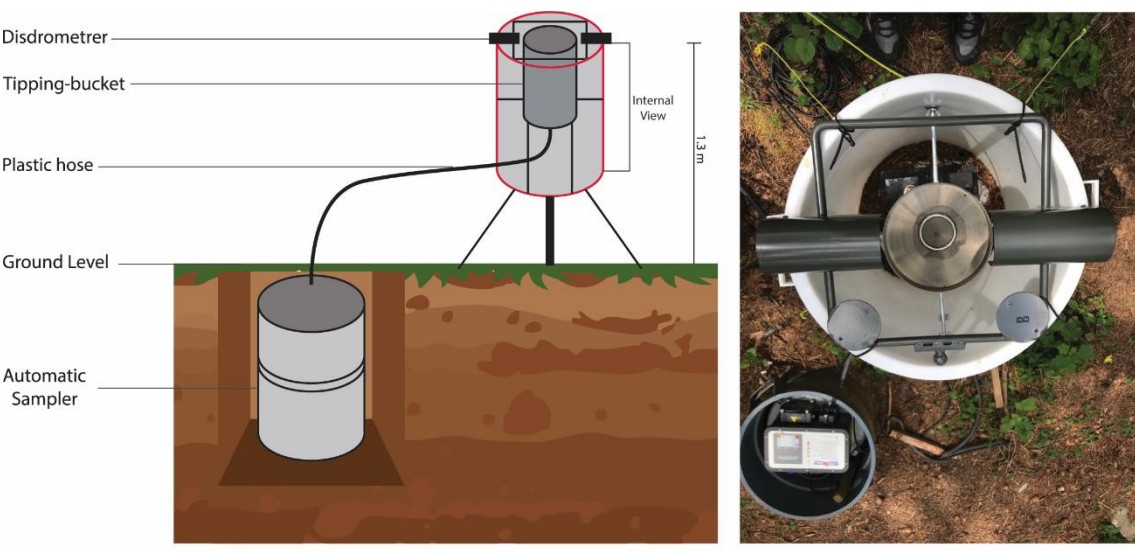

**Figure 2.** Setup for continuous measurement and sampling of open rainfall and throughfall. Diagram of the experimental equipment (left) and top view of the equipment installed in the forest stand (right).









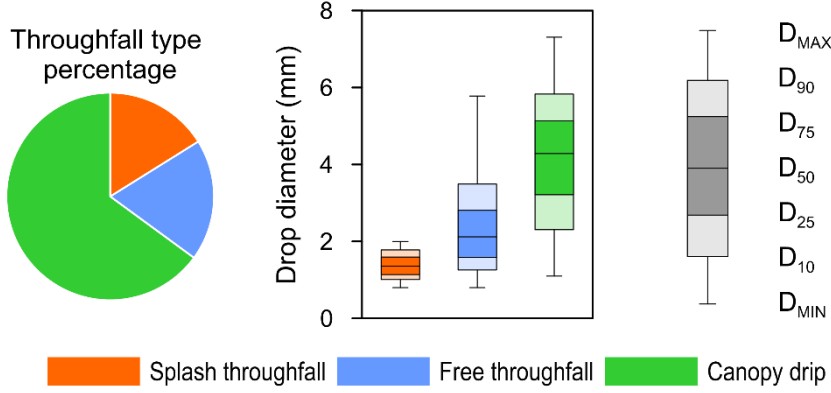

**Figure 3.** Throughfall type percentages (pie chart) and drop diameters for the three throughfall types (boxplots) for the 21 events studied. The drop diameters are based on the mean of event examined.



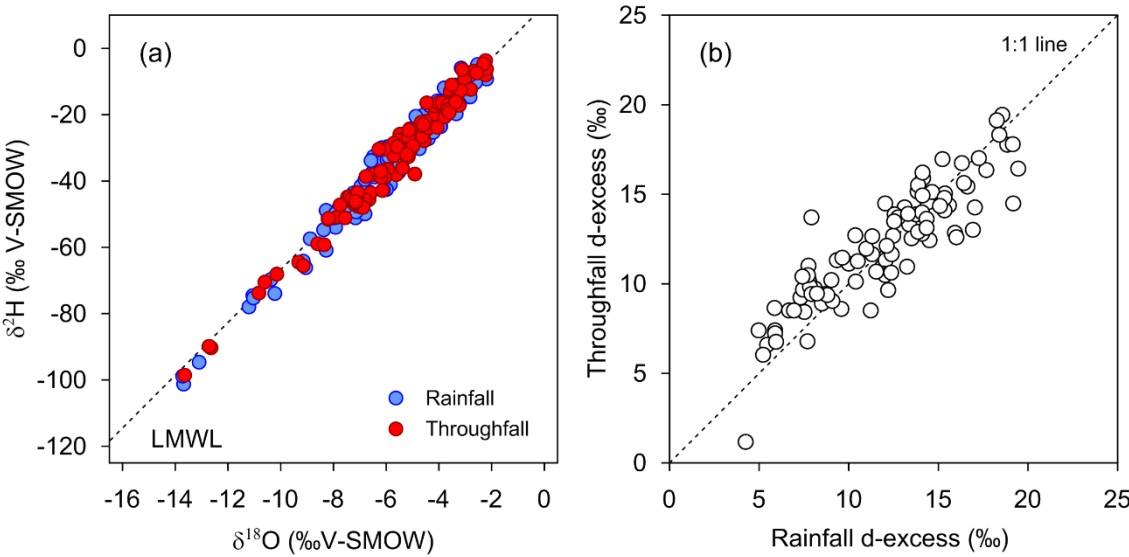

**Figure 4.** (a) $\delta^{18}O$ and $\delta^{2}H$ values of open rainfall and throughfall. The dashed line shows the local meteoric water line (LMWL). (b) the relationship between the deuterium excess ($d$-excess) of open rainfall and throughfall.

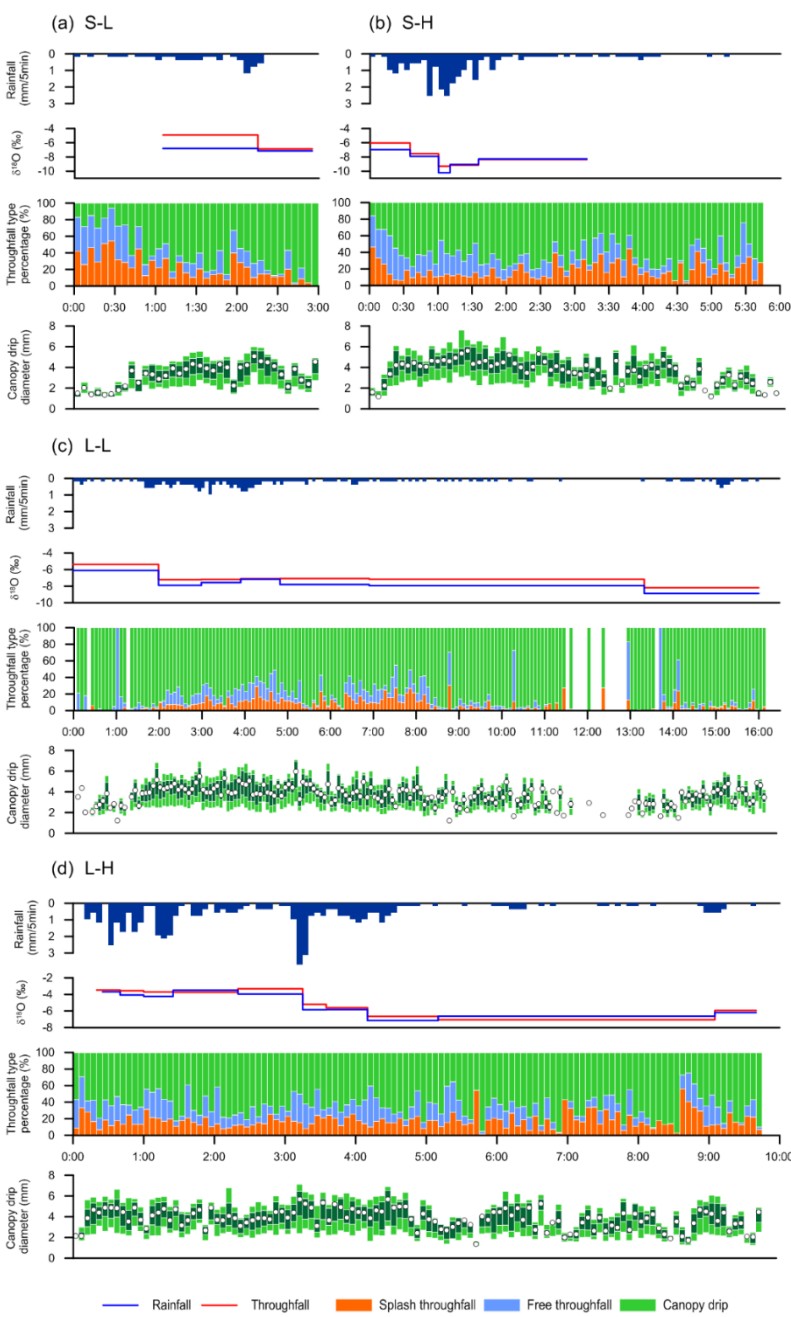


**Figure 5.** Temporal evolution of rainfall and throughfall isotopic composition ($\delta^{18}$O), throughfall type percentages based on volume, and drop diameter of canopy drip for: a) short duration-low intensity event (S-L), b) short duration-high intensity event (S-H), c) long duration-low intensity event (L-L), and d) long duration-high intensity event (L-H). The drop diameter is shown in boxplots with respective cumulative drop volume percentiles (light green: 10% and 90%, dark green: 25% and

75%, white dot: 50%).



**Figure 6.** Boxplot of the isotopic shift ($\delta^{18}O_{TF-RF}$) *versus* classes of: (a) differences in median volume drop diameter ($D_{50\ TF-RF}$), (b) cumulative rainfall throughout the rainfall event, (c) sampling time (i.e. time each 5 mm sample took to be filled), and (d) rainfall kinetic energy per sample. Light blue dots represent outliers.



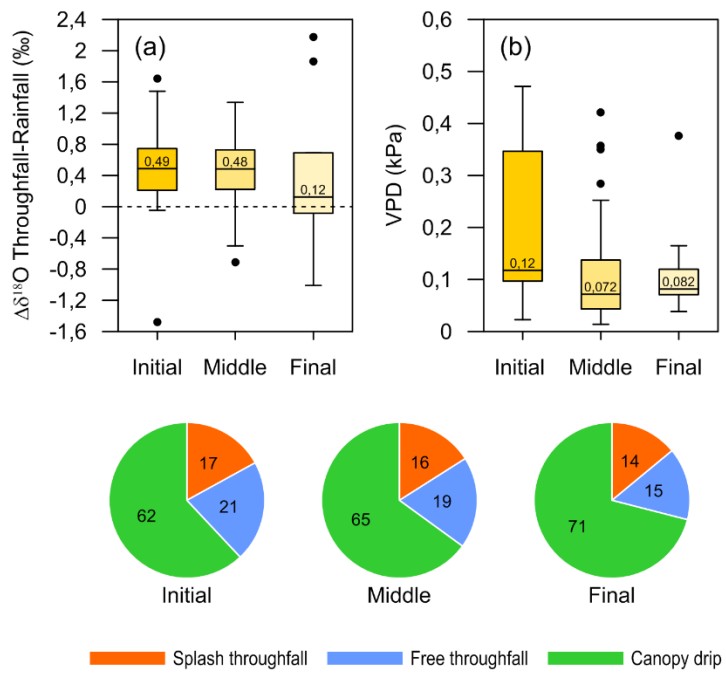


**Figure 7.** Boxplot of the intra-event dynamic observed in the initial, middle and final stages of 16 rainfall events (> 10 mm) for a) isotopic shift differences ($\delta^{18}O_{TF-RF}$), and b) VPD. Black dots represent outliers. Pie charts shows the proportion of throughfall types (%) for the different stages of rainfall events.







## Appendix A


**Constraints with Arduino datalogging systems**

Unlike the logging system by laptop with A/D convertor (Nanko et al., 2006; Levia et al., 2019), the Arduino system could not record all the temporal variation of output voltage due to the time required to record the data onto the SD card and an

insufficient memory (compared to the random access memory –RAM– used in common devices). When a drop passes through the laser beam, output voltage was collected as shown in Fig. A1. When more than three continuous data values of output voltage were less than the threshold voltage (= base voltage * 0.98, here), the Arduino system recorded five values, i.e. the base voltage, the minimum output voltage, the first output voltage, the last output voltage and the number of data between the first and last output voltage, by a signal of drop data and a time stamp.

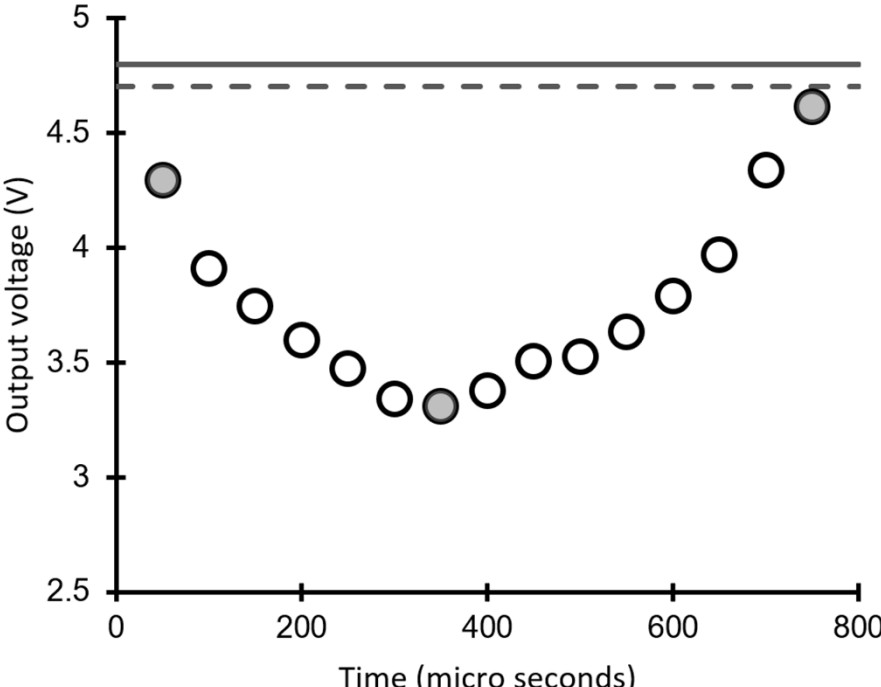


**Figure A1.** An example of temporal variation of output voltage by a drop passed through the laser beam. The solid line denotes base voltage, the dashed line denotes threshold voltage, the open circle denotes output voltage every 50 microseconds and the closed circle denotes the data recording onto the Arduino.

The Arduino threshold voltage setting means that drops with diameter < 0.8 mm could not be measured by this system. The Arduino system also has the disadvantage of failing to account for all drops when rainfall or throughfall continuously pass through the laser beam, due to its limited sampling speed. Finally, when several drops simultaneously pass in parallel in the



same direction of the laser beam, only the nearest drop to the transmitter is recorded. Consequently, due to the limitations of the Arduino datalogging system, on average 39% less incident rainfall and 45% less throughfall was observed between the tipping-bucket and the disdrometer. Fortunately, these differences followed a consistent linear fit. Two assumptions were made for throughfall drop calculations to remedy the deficiencies of the Arduino system: (1) the percentage of throughfall drops < 0.8 mm is a marginal volume; and (2) the throughfall drops not measured by the disdrometers due to the other constraints of the Arduino system were distributed equally among the throughfall types.