# Peer review of "Throughfall isotopic composition in relation to drop size at the intraevent scale in a Mediterranean Scots pine stand"

_Hydrology and Earth System Sciences, 2020_

## Referee Comment (RC1) · Anonymous Referee #1 · 18 Jun 2020

General comments

In the manuscript entitled 'Throughfall isotopic composition in relation to drop size at the intra-event scale in a Mediterranean Scots pine stand' by J. Pinos et al., a study is presented that investigates the relationship between the isotopic composition of throughfall and throughfall drop size - assuming that splash droplet evaporation plays an important role.

For throughfall and rainfall sampling, two setups were installed in 100m distance. Typical meteorological parameters were logged every 30s. 21 summer events were classified by duration and intensity and the isotopic composition of event-based rainfall and

throughfall samples was analysed. Tipping buckets were combined with laser disdrometers for drop size distributions (DSD). Throughfall proportions of the three types free throughfall, canopy drip and splash throughfall were calculated and the interrelationship of isotopic shift and drop size between rainfall and throughfall was analysed.

It could be shown that throughfall is characterised by a lower number of drops, slower drop velocity and larger drop diameter than open rainfall. It could also be shown that the difference in the isotopic composition of throughfall and rainfall increased with a larger proportion of splash droplets and that a higher contribution of splash throughfall and higher VPD results in a greater isotopic shift. Eventually, one may consider changing the manuscript into a technical note, since many of the hydrological aspects are discussed rather briefly and the key contribution is the investigation of the interrelationship between throughfall isotopic composition and throughfall drop size. No mechanistic understanding is provided by the manuscript. The main conclusion also starts with the technological aspect.

Throughfall is typically sampled with multiple samplers to account for the high spatial variability. In this study, only one sampler for throughfall and one for open rainfall was used, which makes the study a bit weak. Some technical issue with the Arduino datalogging system could also be solved to improve the quality of the dataset. Nevertheless, the interrelationship between throughfall isotopic composition and throughfall drop size is a promising new approach to gain insight in small-scale evaporation processes.

Overall, the manuscript is well structured and nicely written. The topic fits well to the scope of the journal and appears to be of interest for the readers; besides the eventual change into a technical note I only suggest minor revisions prior to acceptance and publication in Hydrology and Earth System Sciences.

Global changes

Throughout the manuscript: Please add city and country to the suppliers of the instruments/ parts (ISCO, DT85, Picarro, . . .)

Specific + technical comments

L. 34

Please add Isotopic shifts in throughfall Isotopic shifts are mainly caused. . .

L. 35

please insert "but also by sub-canopy water recycling i.e. evapotranspiration and re-condensation (Green et al., 2015)" after (Allen et al., 2017)

L. 140

. . ."by the". . . instead of . . ."with". . .

L. 164ff and Fig. 3

Please clarify and rephrase: How can "the maximum splash throughfall diameter be set at 2mm" when the threshold for splash throughfall is < 1 mm?

L. 190

Please change to global meteoric water line (GMWL)

L. 361 and L. 374

please delete the "Delta" of the "Delta delta 18-O isotopic shift (. . ."

L. 371

I suggest "," after "meta-analysis"

L. 386 and L. 400

Please delete either ";" or "and" in these sentences.

L. 419

". . .isotopic shift (. . ." should probably be your "Delta delta 18-OTF-RF" in the brackets.

L. 650

events

L. 696, Fig. 5

legend and axis labeling is too small.

References

Green, M. B., Laursen, B. K., Campbell, J. L., McGuire, K. J., and Kelsey, E. P.: Stable water isotopes suggest sub-canopy water recycling in a northern forested catchment, Hydrol Process 29, 5193–5202, doi: 10.1002/hyp.10706, 2015

Please also note the supplement to this comment:
https://www.hydrol-earth-syst-sci-discuss.net/hess-2020-182/hess-2020-182-RC1-supplement.pdf

———————————————

---

## Referee Comment (RC2) · Anonymous Referee #2 · 24 Jun 2020

General Comments: This article investigates the relationship between the stable water isotopic composition of throughfall relative to drop size. The article is well-written and has practical implications for understanding the evolution of isotopic composition as it moves through the forest canopy. The strengths of this manuscript are the high temporal scale at which the measurements were taken and the number of events which were sampled. The weakness is the single throughfall sampler. However, in this way, any variation in the measurements could be attributed to storm characteristics and not to variation among trees. To this end, I think the manuscript is of interest to the HESS readership and could be accepted following minor revisions.

[Figure]

Specific Comments: Line 95-97: Please cite the data source for the climatic data. Line 107: Why were the distances of 0.82 and 1.15 m selected? How was the individual tree selected? Equation 2: OPi was not defined. Can you explain why the assumption of "p is the maximum value under the condition (Fri-pOPi)>0" works? Line 164-165: How can splash throughfall be drops with diameter < 1 mm but the maximum splash diameter is 2 mm? Section 2.4: What time step were the samplers programmed to collect water? Section 2.2 says the tipping buckets recorded every 5 minutes, but were the water samples partitioned into separate collectors for isotopic analysis every 5 minutes too? Figure 5 seems to show isotopic data at non-standard intervals during each storm. Line 209: Provide percent partitioning of max throughfall 48.3 mm event in parentheses. Lines 241-259: Both of these paragraphs could be improved by adding in quantitative data of the % differences. For instance, how much lower was the free throughfall in long duration-low intensity rainfall events? They could also be improved with figures or tables summarizing the data presented. Line 287: The 6 hour drying time will probably evaporate all the water stored on leaf surfaces, but there is almost certainly pre-event water stored in bark tissue that could mix/exchange with the next event. Please address this possibility in the text. Line 305-309: I'm not clear on what the authors are explaining here. Why would there be pre-event water in the sample bottle? Can the authors also remind the reader in the text what the time-step was at which the first and second samples were collected? Line 364: What are the multiple factors/variables? Lin 368/Fig 6a: Are all the datapoints in the first boxplot (<0) of values between -1 and 0 (i.e., of similar distance for the bin compared to the other bins)? The sentence prior to this one says "some significant trends were observed". Was the isotopic shift in the <0 bin statistically significant? If so, indicate in the text and on the figures. If not, please remove the word "significant" from the sentence on Line 367. Line 369-370/Fig 6b: Did the isotopic shift decrease with rainfall or did it just become less variable? Line 377/Fig 6d: In line 374-375 you said there was no clear relationship but here you say there was above the threshold of 300 J/m2. Again, can you really say the shift decreased beyond this threshold or did it become less variable?

[Figure]

Line 421-422: Without statistical analysis, it's not appropriate to say these trends were observed in the data. See previous comments.

Technical Corrections: Line 93: Scot pine should be "Scots" pine Line 102: inconsistent number of decimals Line 260: Here the abbreviations "S-L" and "L-L" are used but in most other instances in the manuscript the full description is written out. Pick one format and be consistent. Line 415: avoid using "showed" twice in this sentence Fig 7: "," should be "." in number formatting

---

## Author Comment (AC1) · 8 Jul 2020

**Reply to Referee #1**

**We appreciate the helpful comments of the reviewer. Please, find below in black the comments of the reviewer and in blue font how we will address each comment and suggestion in the revised manuscript.**

Overall, the manuscript is well structured and nicely written. The topic fits well to the scope of the journal and appears to be of interest for the readers; besides the eventual change into a technical note I only suggest minor revisions prior to acceptance and publication in Hydrology and Earth System Sciences.

**Response: We appreciate the overall positive assessment of our work. Following the recommendation and in consideration of the technical aspects of this work we will change the article to a Technical Note.**

Global changes:

Throughout the manuscript: Please add city and country to the suppliers of the instruments/ parts (ISCO, DT85, Picarro, …)

**Response: Thanks for the recommendation; we will add the city and country of the instruments/parts used in the study.**

Specific + technical comments:

L. 34 Please add Isotopic shifts in throughfall Isotopic shifts are mainly caused…

**Response: We will add the suggested changes in the text.**

L. 35 please insert "but also by sub-canopy water recycling i.e. evapotranspiration and recondensation (Green et al., 2015)" after (Allen et al., 2017)

**Response: Following the recommendation, we will add the statement and reference in the text.**

L. 140 …"by the"… instead of …"with"…

**Response: Following the recommendations, we will change "with" to "by the".**

L. 164 and Fig. 3 Please clarify and rephrase: How can "the maximum splash throughfall diameter be set at 2mm" when the threshold for splash throughfall is < 1 mm?

**Response: To clarify this item, we will reword the sentence: "Splash throughfall is smaller than canopy drip. We set the maximum splash throughfall diameter ($D_{MAX\_SP}$) at 2.0 mm and the minimum canopy drip diameter at 1.0 mm, respectively. It indicated**

**throughfall drops with diameter ($d_i$) from 1.0 to 2.0 mm were generated from the mixture of FR, SP, and DR."**

L. 190 Please change to global meteoric water line (GMWL)

**Response: We will add the suggested changes in the text.**

L. 361 and L. 374 please delete the "Delta" of the "Delta delta 18-O isotopic shift (…"

**Response: We will delete the "Delta" in both lines.**

L. 371 I suggest "," after "meta-analysis"

**Response: We will add it.**

L. 386 and L. 400 Please delete either ";" or "and" in these sentences.

**Response: We will delete the ";" in both lines.**

L. 419 "…isotopic shift (…" should probably be your "Delta delta 18-OTF-RF" in the brackets.

**Response: We will fix it.**

L. 650 events

**Response: We will fix it.**

L. 696, Fig. 5 legend and axis labeling is too small.

**Response: Thanks for the observation, we will fix this.**

---

## Author Comment (AC2) · 8 Jul 2020

**Reply to Referee #2**

We appreciate the helpful comments of the reviewer. Please, find below in black the comments of the reviewer and in blue font how we will address each comment and suggestion in the revised manuscript.

General Comments:

This article investigates the relationship between the stable water isotopic composition of throughfall relative to drop size. The article is well-written and has practical implications for understanding the evolution of isotopic composition as it moves through the forest canopy. The strengths of this manuscript are the high temporal scale at which the measurements were taken and the number of events which were sampled. The weakness is the single throughfall sampler. However, in this way, any variation in the measurements could be attributed to storm characteristics and not to variation among trees. To this end, I think the manuscript is of interest to the HESS readership and could be accepted following minor revisions.

Response:  We appreciate the overall positive assessment of our work.

Specific Comments:

Line 95-97: Please cite the data source for the climatic data.

Response: The climatic data was calculated from the meteorological data collected by the authors in the Vallcebre research catchments. The reference Llorens et al. (2018) will be included.

Line 107: Why were the distances of 0.82 and 1.15 m selected? How was the individual tree selected?

Response: We will clarify both questions in the manuscript as follows: "The rainfall monitoring site was located in an open area approximately 100 m from the Scots pine stand where throughfall was monitored (Fig. 1). The study tree is representative of the forest plot and has a canopy projected area large enough to locate the throughfall instruments. Throughfall was monitored at two randomly selected distances (0.8 and 1.2 m) from the bole of the study tree (Table 1)".

Equation 2: OPi was not defined. Can you explain why the assumption of "p is the maximum value under the condition (Fri-pOPi)>0" works?

Response:  Thank you for the comment. We will define $OP_i$ and clarify the assumption as follows: "where $OP_i$ is the class $i$ of open rainfall and $p$ is the free throughfall fraction (dimensionless, from 0 to 1), which is related to canopy openness. Raindrop impact on the canopy and/or wind and turbulence can cause the canopy to sway during rainfall events, triggering dynamic variation in the degree of canopy openness. Because it is difficult (or impossible) to determine actual $p$, an approximation of $p$ was assigned as

the maximum value under the condition ($FR_i - p\ OP_i$) > 0, utilizing the same protocol as Nakaya et al. (2011). This protocol might overestimate *p*.”

Line 164-165: How can splash throughfall be drops with diameter < 1 mm but the maximum splash diameter is 2 mm?

Response:  To clarify this item, we will reword the sentence: “Splash throughfall is smaller than canopy drip. We set the maximum splash throughfall diameter ($D_{MAX\_SP}$) at 2.0 mm and the minimum canopy drip diameter at 1.0 mm, respectively. It indicated throughfall drops with diameter ($d_i$) from 1.0 to 2.0 mm were generated from the mixture of FR, SP, and DR.” (L. 171-173)

Section 2.4: What time step were the samplers programmed to collect water? Section 2.2 says the tipping buckets recorded every 5 minutes, but were the water samples partitioned into separate collectors for isotopic analysis every 5 minutes too?

Response: Thank you for pointing this out. Automatic samplers were set to collect samples every 5 mm of rainfall, whereas the datalogger recorded the tipping-bucket data every 5 minutes. We will clarify this difference between data measured by the tipping-buckets (time) and collected samples (volume) in the manuscript.

Figure 5 seems to show isotopic data at non-standard intervals during each storm.

Response: Isotopic data depicted in Figure 5 correspond to intervals of 5 mm of rainfall. For that reason, there are different time intervals between samples. We will include a clarification in the caption of figure 5.

Line 209: Provide percent partitioning of max throughfall 48.3 mm event in parentheses.

Response:  We will add this information.

Lines 241-259: Both of these paragraphs could be improved by adding in quantitative data of the % differences. For instance, how much lower was the free throughfall in long duration-low intensity rainfall events? They could also be improved with figures or tables summarizing the data presented.

Response: We agree that adding quantitative data of the % differences could be useful to summarize the data we presented. We will add a table with the percentages as part of the Supplementary Material.

Line 287: The 6 hour drying time will probably evaporate all the water stored on leaf surfaces, but there is almost certainly pre-event water stored in bark tissue that could mix/exchange with the next event. Please address this possibility in the text.

Response: We appreciate this suggestion. According to Llorens et al (2014), in the same study area (with oaks) after the rainfall events, the canopy from 3 m above ground to the top was dry after 6 h during the day and 12 h overnight. Taking into account that these drying times are reasonable for the plot studied, we separate the events in this work. We agree that there is a possibility that the tree boles (2-3 m above the ground) will be wet longer. Although this could have an implication for the

**stemflow isotopic composition, we consider that this would not influence the throughfall isotopic composition.**

Line 305-309: I'm not clear on what the authors are explaining here. Why would there be pre-event water in the sample bottle? Can the authors also remind the reader in the text what the time-step was at which the first and second samples were collected?

**Response: The samples were collected every 5 mm of rain, but the bottles of the automatic samplers were collected every week. Therefore, if for example two events occur during a week, it may happen that the water from the last sample of the first event (in case it does not reach 5 mm) mixes with the water from the first sample of the second event. These mixed samples were discarded from the analysis.**

Line 364: What are the multiple factors/variables?

**Response: Following the reviewer's recommendation the sentence will be improved as follows: "The most likely scenario is that a combination of rainfall characteristics, meteorological variables and isotopic fractionation factors exerted influence on the isotopic fractionation observed in the canopy."**

Lin 368/Fig 6a: Are all the datapoints in the first boxplot (<0) of values between -1 and 0 (i.e., of similar distance for the bin compared to the other bins)? The sentence prior to this one says "some significant trends were observed". Was the isotopic shift in the <0 bin statistically significant? If so, indicate in the text and on the figures. If not, please remove the word "significant" from the sentence on Line 367.

**Response: The first bin in Fig. 6a corresponds to the interval -1.2 to 0. Then, we choose to group it as (<0). Thanks for the indication, we will remove the word "significant".**

Line 369-370/Fig 6b: Did the isotopic shift decrease with rainfall or did it just become less variable?

**Response: Both, the isotopic shift slightly decreased and become less variable with increasing cumulative rainfall. We will clarify the sentence in the manuscript.**

Line 377/Fig 6d: In line 374-375 you said there was no clear relationship but here you say there was above the threshold of 300 J/m2. Again, can you really say the shift decreased beyond this threshold or did it become less variable?

**Response: Following the recommendation, we will delete the incoherent information from the text in Line 377.**

Line 421-422: Without statistical analysis, it's not appropriate to say these trends were observed in the data. See previous comments.

**Response: We agree, and we will modify the sentence to not be interpreted as statistically significant.**

Technical Corrections:

Line 93: Scot pine should be "Scots" pine

**Response: We will fix it.**

Line 102: inconsistent number of decimals

**Response: We will fix it.**

Line 260: Here the abbreviations "S-L" and "L-L" are used but in most other instances in the manuscript the full description is written out. Pick one format and be consistent.

**Response: We checked the manuscript for consistency, and we use the full description in the text and the abbreviations only in brackets.**

Line 415: avoid using "showed" twice in this sentence

**Response: We will fix it.**

Fig 7: "," should be "." in number formatting

**Response: Following the recommendations, we will change "," to "." in Fig. 7.**

References:

Llorens, P., Domingo, F., Garcia-Estringana, P., Muzylo, A., and Gallart, F.: Canopy wetness patterns in a Mediterranean deciduous stand, J. Hydrol., 512, 254-262, https://doi.org/10.1016/j.jhydrol.2014.03.007, 2014.

Llorens, P., Gallart, F., Cayuela, C., Roig-Planasdemunt, M., Casellas, E., Molina, A. J., Moreno-de las Heras, M., Bertran, G., Sánchez-Costa, E., and Latron, J.: What have we learnt about Mediterranean catchment hydrology? 30 years observing hydrological processes in the Vallcebre research catchments, Geogr. Res. Lett., 44, 475-501, https://doi.org/10.18172/cig.3432, 2018.

---

## Author Response (AR1)

**General comments to the Editor and Reviewer**

We thank the Editor Miriam Coenders-Gerrits and both Reviewers for their time to provide critical feedback to our manuscript. The article has clearly benefited from the helpful suggestions of the two anonymous referees. In accordance with recommendation of the handling Editor, we have decided to continue with the manuscript submission as a research article. Along with both reviewers, we agree that the results are promising and that further work is needed to confirm or expand upon our findings. Further work will provide even greater insights into the interrelationships between throughfall partitioning and isotopic shift that we have begun to uncover in this study. Finally, the data are available upon request to the corresponding author.

Please, find below a complete list of answers to the comments and changes to the paper carried out to carefully address your remarks and the suggestions proposed by the two referees. The comments are shown below in black regular font. Our responses are presented below each comment in blue bold font. A marked-up version of the manuscript showing the specific changes we made is submitted along with this letter.

We hope that this will allow the editor to assess the revision without another round of peer-review, since the requested changes were minor and all remarks of the reviewers were accounted for.

**Referee #1**

Overall, the manuscript is well structured and nicely written. The topic fits well to the scope of the journal and appears to be of interest for the readers; besides the eventual change into a technical note I only suggest minor revisions prior to acceptance and publication in Hydrology and Earth System Sciences.

**Response: We appreciate the overall positive assessment of our work. Despite the initial consideration to change the article to a technical note, following the handling Editor's recommendation, we have decided to continue the manuscript submission as a research article due to the significant albeit preliminary research findings.**

Global changes:

Throughout the manuscript: Please add city and country to the suppliers of the instruments/ parts (ISCO, DT85, Picarro, …)

**Response: Thanks for the recommendation; we have added the city and country of the instruments/parts used in the study.**

Specific + technical comments:

L. 34 Please add Isotopic shifts in throughfall Isotopic shifts are mainly caused…

**Response: We have added the suggested changes in the text.**

L. 35 please insert "but also by sub-canopy water recycling i.e. evapotranspiration and recondensation (Green et al., 2015)" after (Allen et al., 2017)

**Response: Following the recommendation, we have added the statement and reference in the text.**

L. 140 …"by the"… instead of …"with"…

**Response: Following the recommendation, we have changed "with" to "by the".**

L. 164 and Fig. 3 Please clarify and rephrase: How can "the maximum splash throughfall diameter be set at 2mm" when the threshold for splash throughfall is < 1 mm?

**Response: To clarify this item, we have reworded this sentence: "Splash throughfall is smaller than canopy drip. We set the maximum splash throughfall diameter ($D_{MAX\_SP}$) at 2.0 mm and the minimum canopy drip diameter at 1.0 mm, respectively. It indicated throughfall drops with diameter ($d_i$) from 1.0 to 2.0 mm were generated from the mixture of FR, SP, and DR."**

L. 190 Please change to global meteoric water line (GMWL)

**Response: We have added the suggested changes in the text.**

L. 361 and L. 374 please delete the "Delta" of the "Delta delta 18-O isotopic shift (…"

**Response: We have deleted the "Delta" in both lines.**

L. 371 I suggest "," after "meta-analysis"

**Response: We have added it.**

L. 386 and L. 400 Please delete either ";" or "and" in these sentences.

**Response: We have deleted the ";" in both lines.**

L. 419 "…isotopic shift (…" should probably be your "Delta delta 18-OTF-RF" in the brackets.

**Response: We have fixed it.**

L. 650 events

**Response: We have fixed it.**

L. 696, Fig. 5 legend and axis labeling is too small.

**Response: Thanks for the observation, we have fixed this.**

**Reply to Referee #2**

General Comments:

This article investigates the relationship between the stable water isotopic composition of throughfall relative to drop size. The article is well-written and has practical implications for understanding the evolution of isotopic composition as it moves through the forest canopy. The strengths of this manuscript are the high temporal scale at which the measurements were taken and the number of events which were sampled. The weakness is the single throughfall sampler. However, in this way, any variation in the measurements could be attributed to storm characteristics and not to variation among trees. To this end, I think the manuscript is of interest to the HESS readership and could be accepted following minor revisions.

**Response: We appreciate the overall positive assessment of our work.**

Specific Comments:

Line 95-97: Please cite the data source for the climatic data.

**Response: The climatic data was calculated from the meteorological data collected by the authors in the Vallcebre research catchments. The reference Llorens et al. (2018) was included.**

Line 107: Why were the distances of 0.82 and 1.15 m selected? How was the individual tree selected?

**Response: We have clarified both questions in the manuscript as follows: "The rainfall monitoring site was located in an open area approximately 100 m from the Scots pine stand where throughfall was monitored (Fig. 1). The study tree is representative of the forest plot and has a canopy projected area large enough to locate the throughfall instruments. Throughfall was monitored at two randomly selected distances (0.8 and 1.2 m) from the bole of the study tree (Table 1)".**

Equation 2: OPi was not defined. Can you explain why the assumption of "p is the maximum value under the condition (Fri-pOPi)>0" works?

**Response: Thank you for the comment. We have defined $OP_i$ and clarified the assumption as follows: "where $OP_i$ is the class $i$ of open rainfall and $p$ is the free throughfall fraction (dimensionless, from 0 to 1), which is related to canopy openness.**

**Raindrop impact on the canopy and/or wind and turbulence can cause the canopy to sway during rainfall events, triggering dynamic variation in the degree of canopy openness. Because it is difficult (or impossible) to determine actual $p$, an approximation of $p$ was assigned as the maximum value under the condition ($FR_i - p$ $OP_i$) > 0, utilizing the same protocol as Nakaya et al. (2011). This protocol might overestimate $p$.”**

Line 164-165: How can splash throughfall be drops with diameter < 1 mm but the maximum splash diameter is 2 mm?

**Response: To clarify this item, we have reworded this sentence: “Splash throughfall is smaller than canopy drip. We set the maximum splash throughfall diameter ($D_{MAX\_SP}$) at 2.0 mm and the minimum canopy drip diameter at 1.0 mm, respectively. It indicated throughfall drops with diameter ($d_i$) from 1.0 to 2.0 mm were generated from the mixture of FR, SP, and DR.”**

Section 2.4: What time step were the samplers programmed to collect water? Section 2.2 says the tipping buckets recorded every 5 minutes, but were the water samples partitioned into separate collectors for isotopic analysis every 5 minutes too?

**Response: Thank you for pointing this out. Automatic samplers were set to collect samples every 5 mm of rainfall, whereas the datalogger recorded the tipping-bucket data every 5 minutes. We have clarified this difference between data measured by the tipping-buckets (time) and collected samples (volume) in the manuscript.**

Figure 5 seems to show isotopic data at non-standard intervals during each storm.

**Response: Isotopic data depicted in Figure 5 correspond to intervals of 5 mm of rainfall. For that reason, there are different time intervals between samples. We have included a clarification in the caption of figure 5.**

Line 209: Provide percent partitioning of max throughfall 48.3 mm event in parentheses.

**Response: We have added this information.**

Lines 241-259: Both of these paragraphs could be improved by adding in quantitative data of the % differences. For instance, how much lower was the free throughfall in long duration-low intensity rainfall events? They could also be improved with figures or tables summarizing the data presented.

**Response: We agree that adding quantitative data of the % differences could be useful to summarize the data we presented. We have added a table with the percentages as part of the Supplementary Material.**

Line 287: The 6 hour drying time will probably evaporate all the water stored on leaf surfaces, but there is almost certainly pre-event water stored in bark tissue that could mix/exchange with the next event. Please address this possibility in the text.

**Response: We appreciate this suggestion. According to Llorens et al (2014), in the same study area (with oaks) after the rainfall events, the canopy from 3 m above**

**ground to the top was dry after 6 h during the day and 12 h overnight. Taking into account that these drying times are reasonable for the plot studied, we separate the events in this work. We agree that there is a possibility that the tree boles (2-3 m above the ground) will be wet longer. Although this could have an implication for the stemflow isotopic composition, we consider that this would not influence the throughfall isotopic composition.**

Line 305-309: I'm not clear on what the authors are explaining here. Why would there be pre-event water in the sample bottle? Can the authors also remind the reader in the text what the time-step was at which the first and second samples were collected?

**Response: The samples were collected every 5 mm of rain, but the bottles of the automatic samplers were collected every week. Therefore, if for example two events occur during a week, it may happen that the water from the last sample of the first event (in case it does not reach 5 mm) mixes with the water from the first sample of the second event. These mixed samples were discarded from the analysis.**

Line 364: What are the multiple factors/variables?

**Response: Following the reviewer's recommendation the sentence was improved as follows: "The most likely scenario is that a combination of rainfall characteristics, meteorological variables and isotopic fractionation factors exerted influence on the isotopic fractionation observed in the canopy."**

Lin 368/Fig 6a: Are all the datapoints in the first boxplot (<0) of values between -1 and 0 (i.e., of similar distance for the bin compared to the other bins)? The sentence prior to this one says "some significant trends were observed". Was the isotopic shift in the <0 bin statistically significant? If so, indicate in the text and on the figures. If not, please remove the word "significant" from the sentence on Line 367.

**Response: The first bin in Fig. 6a corresponds to the interval -1.2 to 0. Then, we choose to group it as (<0). Thanks for the indication, we have removed the word "significant".**

Line 369-370/Fig 6b: Did the isotopic shift decrease with rainfall or did it just become less variable?

**Response: Both, the isotopic shift slightly decreased and become less variable with increasing cumulative rainfall. We have clarified the sentence in the manuscript.**

Line 377/Fig 6d: In line 374-375 you said there was no clear relationship but here you say there was above the threshold of 300 J/m2. Again, can you really say the shift decreased beyond this threshold or did it become less variable?

**Response: Following the recommendation, we have deleted the incoherent information from the text in Line 377.**

Line 421-422: Without statistical analysis, it's not appropriate to say these trends were observed in the data. See previous comments.

**Response: We agree, and we have modified the sentence to not be interpreted as statistically significant.**

Technical Corrections:

Line 93: Scot pine should be "Scots" pine

**Response: We have fixed it.**

Line 102: inconsistent number of decimals

**Response: We have fixed it.**

Line 260: Here the abbreviations "S-L" and "L-L" are used but in most other instances in the manuscript the full description is written out. Pick one format and be consistent.

**Response: We checked the manuscript for consistency, and we use the full description in the text and the abbreviations only in brackets.**

Line 415: avoid using "showed" twice in this sentence

**Response: We have fixed it.**

Fig 7: "," should be "." in number formatting

**Response: Following the recommendations, we have changed "," to "." in Fig. 7.**

References:

[revised manuscript text omitted]

---

## Author Response (AR2)

Following the Editor's recommendation, the authors decided to make the data publicly accessible from the CSIC open repository. The "Data availability" statement within the article was changed for this purpose, and the link to the dataset was included.